# Impact of dysfunctional parenting, affective temperaments, and stressful life events on the development of melancholic and non-melancholic depression: A path analysis study

**Yu Tamada**[1,2]*, **Takeshi Inoue**[3], **Atsushi Sekine**[4], **Hiroyuki Toda**[5], **Minoru Takeshima**[3,6], **Masaaki Sasaki**[7], **Yota Fujimura**[1], **Susumu Ohmae**[2]

**1** Department of Psychiatry, Tokyo Medical University Hachioji Medical Center, Hachioji, Tokyo, Japan,
**2** Department of Psychiatry, Toranomon Hospital, Tokyo, Japan, **3** Department of Psychiatry, Tokyo Medical University, Tokyo, Japan, **4** The Medical Foundation of Keishin-Kai, Kyouwa Hospital, Daisen, Akita, Japan, **5** Department of Psychiatry, National Defense Medical College, Tokorozawa, Saitama, Japan, **6** Shibata Hospital, Takaoka, Toyama, Japan, **7** Department of Psychiatry, Toranomon Hospital Kajigaya, Kawasaki, Kanagawa, Japan

\* y-tmd@umin.ac.jp

## Abstract

### Background

The influence of psychosocial factors on differentiating between melancholic depression (MEL) and non-melancholic depression (NMEL) remains unclear. In this study, we aimed to investigate the interrelationship between dysfunctional parenting, personality traits, stressful life events, and the diagnosis of MEL and NMEL among patients with major depressive disorder (MDD).

### Methods

Ninety-eight patients with MDD completed the following self-administered questionnaires: the Parental Bonding Instrument (PBI) for dysfunctional parenting, the short version of the Temperament Evaluation of Memphis, Pisa, Paris and San Diego-autoquestionnaire version (TEMPS-A) for affective temperaments, and the Life Experiences Survey (LES) for stressful life events. The data were analyzed using single and multiple regression analyses and path analysis.

### Results

Dysfunctional parenting did not have a significant direct effect on MEL. However, paternal care had a significant indirect effect on MEL through depressive temperament. The total indirect effect of paternal care on MEL was significant (indirect path coefficient = 0.161, $p$ <0.05). In other words, low levels of paternal care were associated with the development of NMEL via increased depressive temperament. None of the paths from paternal care to MEL via negative change scores of the LES were significant.

**Data availability statement:** Data cannot be shared publicly because of Ethics Committee restriction. Data are available from the Internal Review Board of Toranomon Hospital (Japan) (contact via email: ytmd@tokyo-med.ac.jp) for researchers who meet the criteria for access to confidential data.

**Funding:** This work was supported by a Grant-in-Aid for Early-Career Scientists from the Japan Society for the Promotion of Science (JSPS KAKENHI grant no.: JP19K14435) to YT; and grants from the Okinaka Memorial Institute for Medical Research to YT. The funders had no role in study design, data collection and analysis, decision to publish, or preparation of the manuscript.

**Competing interests:** I have read the journal's policy and the authors of this manuscript have the following competing interests: YT received honoraria from Otsuka Pharmaceutical, Sumitomo Pharma, Eisai, MSD and Meiji Seika Pharma. TI has received personal compensation from Mochida Pharmaceutical, Takeda Pharmaceutical, Eli Lilly, Janssen Pharmaceutical, MSD, Taisho Toyama Pharmaceutical, Yoshitomiyakuhin, and Daiichi Sankyo; grants from Shionogi, Astellas, Tsumura, and Eisai; and grants and personal compensation from Otsuka Pharmaceutical, Sumitomo Pharma, Mitsubishi Tanabe Pharma, Kyowa Pharmaceutical Industry, Pfizer, Novartis Pharma, and Meiji Seika Pharma; and is a member of the advisory boards of Pfizer, Novartis Pharma, and Mitsubishi Tanabe Pharma. AS has received personal compensation from Otsuka Pharmaceutical, Meiji Seika Pharma, Takeda Pharmaceutical, and Sumitomo Pharma. HT has received lecture fees from Sumitomo Pharma, Otsuka Pharmaceutical, Takeda Pharmaceutical, Pfizer, Yoshitomi Pharmaceutical, and Viatris. MT has received lecture fees from Kyowa Pharmaceutical, Otsuka Pharmaceutical, Sumitomo Pharma, Takeda Pharmaceutical, and Lundbeck. MS has received lecture fees from Viatris. YF has received honoraria from Sumitomo Pharma, and research grants from Shionogi and Sumitomo Pharma. SO has received honoraria from Viatris and Takeda Pharmaceutical. This does not alter our adherence to PLOS ONE policies on sharing data and materials.

## Limitations

This study used cross-sectional data, so the possibility that current depressive status may affect the assessment of LES and TEMPS-A cannot be ruled out.

## Conclusions

We found that low levels of paternal care did not directly affect the development of NMEL, but affected the development of NMEL through the mediation of depressive temperament rather than stressful life events.

## Introduction

Dysfunctional parenting, personality traits, and stressful life events have been widely studied as psychosocial factors that affect the onset of major depressive disorder (MDD). Regarding the association between dysfunctional parenting and MDD, it has been reported that parental "affectionless control", i.e., lack of care and high degree of overprotection, are associated with MDD [1–4]. Among personality traits, neuroticism is known to be associated with MDD [5, 6]. According to a meta-analysis by Solmi et al. [7], patients with MDD have higher depressive, cyclothymic, irritable, and anxious temperaments, and a lower hyperthymic temperament than healthy individuals. Both causal and non-causal associations have also been reported between the development of MDD and stressful life events [8].

However, because MDD patients have been described as a heterogenous group [9], investigating factors associated with the development of depressive disorders in patients with specific subtypes of MDD may provide more accurate results. The most widely studied topic of MDD subtypes is the distinction between melancholic depression (MEL) and non-melancholic depression (NMEL). Traditionally, MEL has been referred to as endogenous depression, and NMEL as neurotic or reactive depression [10]. MEL is a subtype of MDD with stereotypic symptomatology, such as psychomotor disturbance and anhedonia, which is considered to respond preferentially to physical therapies including pharmacotherapy, and is assumed to have a biological basis [11]. On the other hand, NMEL has been associated with a more vulnerable personality styles than MEL [12], and has been suggested to be associated with having experienced greater difficulties with their parents during childhood and adolescence [13].

A comparison of the effects of dysfunctional parenting, NMEL patients were more likely to have experienced a lack of parental care and higher degree of overprotection than MEL patients [14, 15]. Lower levels of care were associated more closely with NMEL than overprotection [16]. It should be noted, however, that in these studies [14–16], the diagnosis of MEL and NMEL were made by clinical judgment of the psychiatrists, and were not based on specific diagnostic criteria. There are multiple diagnostic criteria for MEL, and the results may vary depending on the criteria used. For example, using the criteria for MEL of DSM-III-R and DSM-IV, there was no significant difference in the degree of dysfunctional parenting experienced between patients with MEL and those with NMEL [17].

Traditionally, MEL patients have been considered to have less personality disorders than NMEL patients [18, 19]. However, MEL patients reportedly have a certain amount of personality deviation compared with healthy controls [20]. MEL patients as defined by DSM-IV have higher irritability, lower cooperativeness [21], and higher introversion [22] than NMEL patients. On the other hand, NMEL patients as defined by DSM-III have higher interpersonal sensitivity than MEL patients [12, 23]. In a study of affective temperaments, NMEL patients as

defined by DSM-5 had significantly higher depressive temperament than MEL patients (our unpublished data).

Regarding the association with life events, there are many reports that MEL patients are less affected by stressful life events than NMEL patients [24, 25], although this association is not consistently supported [26]. Studies in the 1990s indicated that the impact of life events might vary depending on the number of depressive episodes [27]. In other words, both MEL and NMEL patients experienced the same severity of stressful life events prior to the onset of their first depressive episode. For the second and subsequent depressive episodes, there was a trend toward fewer severe stressful life events in MEL patients and no change in stressful life event severity in NMEL patients compared with the first episode. However, Mitchell et al. [26] reported different results. The effects of genetic predispositions that tend to induce adverse life events has also been noted [28], and it is possible that an individual's biological predispositions and life events are not independent. These results indicate that there is a complicated association between life events and depressive disorders [25].

A significant temporal gap exists between the experience of dysfunctional parenting during childhood and the onset of MDD in adulthood. It is natural to assume that there is a complex association between the onset of MDD and dysfunctional parenting, personality traits, and stressful life events. Their interrelationship needs to be examined to analyze the psychosocial factors of depressive disorders. A study of MDD patients showed that dysfunctional parenting affects the severity of depression via personality traits such as "perfectionism" and "concern over mistakes" [29]. In addition, a study of adults in the general population that used structural equation modeling to analyze the interrelationship of depression with neuroticism, dysfunctional parenting and stressful life events showed that low parental care and high overprotection increased depressive symptoms via increased neuroticism, but not via negative life events [30].

Few studies have investigated the interrelationship between the development of MEL and dysfunctional parenting, personality traits, and stressful life events. Carter et al. [31] showed that low care was associated with an increased incidence of personality disorders and that low care was not associated with MEL, but they did not investigate the interrelationship between parenting, personality disorders, and MEL. Whiffen et al. [32] used path analysis to show that a lack of parental care is associated with the development of NMEL via personality dysfunction characterized by self-criticism. However, this study did not investigate the associations of NMEL with stressful life events. We hypothesized that dysfunctional parenting affects the development of NMEL via stressful life events and affective temperaments. In this study, we aimed to clarify the interrelationship between dysfunctional parenting, affective temperaments, stressful life events, and the diagnosis of MEL and NMEL using path analysis.

## Materials and methods

### Subjects

The study included 106 patients diagnosed as having MDD according to DSM-5 [5]. This group included both inpatients and outpatients and was the same cohort as in our previous study [33]. The patients were recruited from January 2018 to March 2019. All patients were diagnosed, evaluated, and treated by psychiatrists with at least five years of clinical experience. Authors had not access to information that could identify individual participants after data collection. Eligible patients had to meet the following criteria: (a) meet the diagnostic criteria for MDD, (b) be between 20 and 69 years of age, and (c) have sufficient capacity to provide informed consent. Exclusion criteria included: (a) presence of serious physical illness, (b) presence of organic mental illness, and (c) serious suicidal ideation. Data from 98 of the 106

participants who fully completed all self-administered questionnaires were included in the analysis. The diagnosis of MEL was determined based on the criteria of the DSM-5 specifier "with melancholic features".

Written consent was provided by all participants to participate in the study. This study was conducted in accordance with the Helsinki Declaration and was approved by the Institutional Review Board of Toranomon Hospital (study approval no. 1516-H).

### Assessment

The following demographic data were evaluated for all MDD patients: age, sex, employment status, years of education, marital status, number of offspring, living alone, presence of physical illness, presence of comorbid mental disorders, and family history of mood disorders in first-degree relatives. In addition, the following clinical characteristics of MDD were evaluated: age at onset of the first depressive episode, duration of illness, number of depressive episodes, presence of relapse, history of suicide attempts, and treatment resistance. The definition of treatment resistance was the absence of a response to at least 2 antidepressants from different pharmacological classes that were administered for at least 4 consecutive weeks. All subjects completed the following 4 questionnaires at the time of study enrollment.

### Patient Health Questionnaire-9 (PHQ-9)

The PHQ-9 is a self-administered questionnaire developed by Spitzer et al [34] as a screening tool for depressive symptoms and consists of 9 items that correspond to the diagnostic criteria for MDD. This scale can also be used as an indicator of the severity of depressive disorders [35]. In this study, we used the total score of the Japanese version of the PHQ-9 [36] as an indicator of the severity of depressive symptoms. Each item was scored between 0 to 3 points, resulting in a total score ranging from 0 to 27 points.

### Parental Bonding Instrument (PBI)

The PBI is a self-administered questionnaire developed by Parker [37] to assess parenting experiences until the age of 16. Participants evaluate both their father's and mother's parenting attitudes across 25 items. Parker [37] argued that there are two dimensions to parenting attitudes: care-neglect and overprotection-autonomy. Of the 25 items, 12 reflect scores for care, and 13 reflect scores for overprotection. The rating for each item is on a scale of 0 to 3, with some items being reverse-scored. The overall care score can range from 0 to 36, whereas the total overprotection score can range from 0 to 39. In this study, the total care and overprotection scores for each participant's father and mother, as evaluated using the Japanese version of the PBI [38], were used in the analysis.

### Short version of the Temperament Evaluation of Memphis, Pisa, Paris and San Diego-autoquestionnaire version (TEMPS-A)

The TEMPS-A is a self-administered questionnaire developed by Akiskal et al. [39] to assess subtypes of affective temperament, and its short version consists of 39 items. Participants rated each item as a yes or no (yes = 2 points, no = 1 point). The Japanese version of the TEMPS-A was translated by Matsumoto et al. [40], and has been confirmed to have validity and reliability. In this study, the average score for each of the five temperament subtypes (depressive, cyclothymic, irritable, anxious, and hyperthymic) was evaluated dimensionally.

### Life Experiences Survey (LES)

The LES is a self-administered questionnaire developed by Sarason et al. [41] to assess whether individuals have experienced life events that brought changes to their lives in the past year. The questionnaire consists of 57 items, and participants rate whether they were positively or negatively affected by each event on a seven-point scale ranging from "extremely positive (+3)" to "extremely negative (–3)". The "positive change score" is the absolute value of the total score for events rated as having a positive impact, and the "negative change score" is the absolute value of the total score for events rated as having a negative impact. The Japanese version of the LES [42] was used in this study.

### Data analysis

First, demographic and clinical features were compared between MEL and NMEL patients. The chi-square test, and if necessary, the Fisher's exact test were used to analyze categorical data, and the unpaired $t$-test was used for continuous data. Second, a logistic regression analysis was performed with the diagnosis of MEL or NMEL as the dependent variable, and the scores of the five temperament subtypes of the TEMPS-A as independent variables. Furthermore, logistic regression analysis was performed with the diagnosis of MEL or NMEL as the dependent variable, and the scores of the four subscales of the PBI as independent variables. In both analyses, the forced entry method was used. Third, Pearson's correlation analysis was performed to investigate the association between scores on the five temperament subtypes of the TEMPS-A, the four subitems of the PBI, and the two subitems of the LES. Fourth, multiple regression analyses were conducted with the scores of the TEMPS-A subtypes as the dependent variable and the scores of the four subitems of the PBI as the independent variables for each of the five affective temperaments. The forced entry method was used for the analysis. Finally, based on these results, a path analysis model was created to conduct mediation analysis with the diagnosis of MEL or NMEL as the dependent variable. In the path model, all coefficients were standardized (ranging from –1 to 1). The model fit was evaluated using indices of the root mean square error of approximation (RMSEA), comparative fit index (CFI), and Tucker-Lewis Index (TLI). In accordance with the conventional criteria, an RMSEA less than 0.05, a CFI greater than 0.97, and a TLI greater 0.97 are considered to indicate a good model fit [43]. No correction for multiple testing was performed.

Path analysis was analyzed using Mplus 8.5 (Muthén & Muthén, Los Angeles, CA, USA); all other analyses were performed using SPSS Statistics version 28 (IBM, Armonk, NY, USA). A $p$-value of less than 0.05 was considered to indicate a statistically significant difference between groups.

## Results

### Demographic and clinical characteristics, TEMPS-A, PBI, and LES scores (Table 1)

Comorbidity of psychiatric disorders and family history of mood disorders in first-degree relatives were significantly more common in NMEL patients than in MEL patients. Comorbid psychiatric disorders included panic disorder in 1 patient in the MEL group, and autism spectrum disorder in 3 patients, and social anxiety disorder, generalized anxiety disorder, panic disorder, and borderline personality disorder in 1 patient each in the NMEL group. On the TEMPS-A subscales, depressive temperament scores were significantly higher in NMEL patients than in MEL patients. The PBI subscales did not differ between the two groups. In the LES, the negative change score was significantly higher in NMEL patients than in MEL patients.

**Table 1. Demographic and clinical characteristics, and TEMPS-A, PBI, and LES scores of the patients analyzed in this study.**

| | MEL (*n* = 48) | NMEL (*n* = 50) | *p*-value |
|---|---|---|---|
| Demographics | | | |
| Age, years: mean (S.D.) | 50.8 (13.9) | 48.4 (12.8) | 0.38 |
| Sex (male): *n* (%) | 25 (52.1) | 25 (50.0) | 0.84 |
| Education, years: mean (S.D.) | 14.3 (2.0) | 13.9 (2.2) | 0.41 |
| Marital status (married): *n* (%) | 35 (73.0) | 31 (62.0) | 0.25 |
| Number of offspring | 1.4 (0.92) | 1.3 (1.2) | 0.60 |
| Employment status (employed): *n* (%) | 23 (47.9) | 33 (66.0) | 0.07 |
| Living alone: *n* (%) | 5 (10.4) | 11 (22.0) | 0.12 |
| Comorbid physical disease: *n* (%) | 17 (35.4) | 18 (36.0) | 0.95 |
| Comorbid psychiatric disorder: *n* (%) | 1 (2.1) | 7 (14.0) | 0.03 |
| Treatment setting (inpatients): *n* (%) | 8 (16.7) | 5 (10.0) | 0.33 |
| First-degree relative with a mood disorder: *n* (%) | 7 (14.6) | 17 (34.0) | 0.03 |
| Clinical features | | | |
| Age of first episode: years (S.D.) | 46.4 (13.9) | 41.8 (13.7) | 0.11 |
| Duration of MDD: years (S.D.) | 4.4 (5.7) | 6.6 (8.1) | 0.13 |
| Number of depressive episodes (S.D.) | 2.0 (1.5) | 2.4 (1.6) | 0.33 |
| Previous suicide attempt: *n* (%) | 4 (8.3) | 3 (6.0) | 0.48 |
| Treatment resistance: *n* (%) | 7 (14.6) | 11 (22.0) | 0.34 |
| PHQ-9 score: mean (S.D.) | 10.3 (7.9) | 11.6 (6.5) | 0.36 |
| PBI scores | | | |
| Paternal Care: mean (S.D.) | 21.0 (8.5) | 20.4 (8.9) | 0.70 |
| Paternal Overprotection: mean (S.D.) | 10.0 (7.1) | 11.7 (8.1) | 0.28 |
| Maternal Care: mean (S.D.) | 26.4 (7.2) | 23.9 (9.5) | 0.14 |
| Maternal Overprotection: mean (S.D.) | 9.9 (7.4) | 11.7 (6.6) | 0.21 |
| TEMPS-A scores | | | |
| Depressive score: mean (S.D.) | 1.26 (0.24) | 1.44 (0.30) | < 0.01 |
| Cyclothymic score: mean (S.D.) | 1.24 (0.24) | 1.28 (0.22) | 0.37 |
| Hyperthymic score: mean (S.D.) | 1.17 (0.23) | 1.18 (0.25) | 0.99 |
| Irritable score: mean (S.D.) | 1.10 (0.18) | 1.13 (0.18) | 0.52 |
| Anxious score: mean (S.D.) | 1.28 (0.37) | 1.41 (0.37) | 0.08 |
| LES scores | | | |
| Positive change score: mean (S.D.) | 1.0 (1.9) | 1.3 (2.5) | 0.55 |
| Negative change score: mean (S.D.) | 6.3 (6.9) | 10.8 (8.2) | < 0.01 |

Continuous data were analyzed by the unpaired *t*-test, and categorical data were analyzed by the χ² test, and when appropriate, with the Fisher's exact test.

Abbreviations: LES, Life Experiences Survey; MDD, major depressive disorder; MEL, melancholic major depressive disorder; NMEL, non-melancholic major depressive disorder; PBI, Parental Bonding Instrument; PHQ-9, Patient Health Questionnaire-9; S.D., standard deviation; TEMPS-A, Temperament Evaluation of Memphis, Pisa, Paris and San Diego-autoquestionnaire

## Logistic regression analysis

The results of logistic regression analysis with the diagnosis of MEL or NMEL as the dependent variable, and the five TEMPS-A subscale scores as independent variables are shown in Table 2. Only a low depressive temperament score was identified as a statistically significant feature distinguishing NMEL patients from MEL patients. The predictive accuracy rate was 61.2%. Logistic regression analysis with the diagnosis of MEL or NMEL as the dependent

**Table 2. Results of logistic regression analysis with MEL/ NMEL as the dependent variable and TEMPS-A scores as independent variables.**

| Variable | Forced entry method | | | | |
|---|---|---|---|---|---|
| | B | S.E. | p-value | OR | 95%CI |
| Depressive temperament | −2.77 | 1.05 | < 0.01 | 0.06 | 0.01–0.49 |
| Cyclothymic temperament | 0.98 | 1.18 | 0.41 | 2.65 | 0.26–26.99 |
| Hyperthymic temperament | −0.42 | 0.99 | 0.67 | 0.66 | 0.09–4.57 |
| Irritable temperament | 0.91 | 1.41 | 0.52 | 2.49 | 0.16–39.24 |
| Anxious temperament | −0.63 | 0.61 | 0.30 | 0.53 | 0.16–1.76 |
| Constant | 2.77 | 1.79 | 0.12 | 16.01 | |

Fit index of this model: $\chi^2$ = 11.60 (p-value < 0.05), Cox-Snell $R^2$ = 0.11, Hosmer-Lemeshow test $p$ = 0.620, predictive accuracy = 61.2%

Dependent variable: the diagnosis of MEL (1) and NMEL (0)

Independent variables: scores of five subscales of the TEMPS-A

Abbreviations: B, partial regression coefficient; CI, confidence interval; MEL, melancholic major depressive disorder; NMEL, non-melancholic major depressive disorder; OR, odds ratio; S.E., standard error

variable, and the four subscale scores of the PBI as independent variables did not identify a significant independent variable that distinguished MEL patients from NMEL patients (data not shown).

## Correlation between TEMPS-A, PBI, and LES scores

The correlations between TEMPS-A subscale scores, PBI subscale scores, and LES subscale scores are shown in Table 3. The scores for depressive temperament showed a negative correlation with paternal care, and a positive correlation with both paternal and maternal overprotection. Additionally, there was a positive correlation between the depressive temperament scores and the negative change score of the LES. Cyclothymic temperament scores showed a negative correlation with both paternal and maternal care, and a positive correlation with paternal overprotection. Furthermore, the cyclothymic temperament scores positively

**Table 3. Correlation (r) between TEMPS-A, PBI, and LES scores.**

| | dep | cyc | hyp | irr | anx | pca | mca | pop | mop | lesp |
|---|---|---|---|---|---|---|---|---|---|---|
| dep | | | | | | | | | | |
| cyc | 0.49** | | | | | | | | | |
| hyp | 0.04 | 0.35** | | | | | | | | |
| irr | 0.42** | 0.32** | 0.25* | | | | | | | |
| anx | 0.30** | 0.24* | −0.07 | 0.07 | | | | | | |
| pca | −0.40** | −0.28** | −0.18 | −0.28** | 0.10 | | | | | |
| mca | −0.15 | −0.24* | −0.24* | −0.26* | 0.11 | 0.44** | | | | |
| pop | 0.35** | 0.25* | 0.14 | 0.21* | 0.03 | −0.60** | −0.44** | | | |
| mop | 0.26* | 0.18 | 0.16 | 0.30** | 0.01 | −0.34** | −0.68** | 0.60** | | |
| lesp | 0.04 | 0.01 | 0.33** | 0.15 | −0.02 | −0.11 | −0.26* | −0.06 | 0.10 | |
| lesn | 0.39** | 0.26** | 0.22* | 0.08 | 0.30** | −0.13 | 0.01 | 0.24* | 0.02 | 0.07 |

$r$ = Pearson's correlation coefficient.

*$p$ < 0.05

**$p$ < 0.01

Abbreviations: anx, anxious temperament; cyc, cyclothymic temperament; dep, depressive temperament; hyp, hyperthymic temperament; irr, irritable temperament; lesn, negative change score of Life Experiences Survey; lesp, positive change score of Life Experiences Survey; mca, maternal care; mop, maternal overprotection; pca, paternal care; pop, paternal overprotection

correlated with the negative change score of the LES. Hyperthymic temperament scores were negatively correlated with maternal care, and positively correlated with both the positive change score and the negative change score of the LES. Irritable temperament scores were negatively correlated with both paternal and maternal care, whereas they were positively correlated with both paternal and maternal overprotection. Lastly, anxious temperament scores demonstrated a positive correlation with negative change score of the LES.

## Multiple regression analysis with TEMPS-A scores as dependent variables and PBI scores as independent variables

Table 4 shows the results of multiple regression analysis, in which PBI subscale scores were used as the independent variable, and TEMPS-A subscale scores were used as the dependent variable. In all cases, the variance inflation factor (VIF) was less than 2.5, and no multicollinearity was observed. Multiple regression analysis with depressive temperament, cyclothymic temperament, and irritable temperament as dependent variables yielded multiple regression equations with $p < 0.05$. However, the model with cyclothymic and irritable temperament as dependent variables was not considered clinically meaningful, because there was no PBI subscale with a partial regression coefficient of $p < 0.05$, and the adjusted $R^2$ was less than 0.1. When depressive temperament scores were the dependent variable, only paternal care scores significantly predicted depressive temperament scores.

## Path analysis

Path analysis was performed to analyze the association between dysfunctional parenting and affective temperaments, stressful life events, and the diagnosis of MEL or NMEL. Depressive temperament was determined to be the only affective temperament included in the path analysis model based on the comparison of mean values between the MEL and the NMEL group (Table 1), the results of logistic regression analysis (Table 2), and the results of multiple regression analysis (Table 4). The negative change score of LES was also included in the path analysis model. This was based on a comparison of means between the MEL and NMEL groups (Table 1) and its correlation with depressive temperament scores (Table 3). Path diagrams of the direct and indirect effects were created and analyzed for each of the four subscales of the PBI.

The results of the standardized path coefficients are shown in Fig 1. All models showed a good fit with an RMSEA of 0.000, CFI of 1.000, and TLI of 1.000.

**Table 4.  Multiple regression analysis with TEMPS-A scores as dependent variables and PBI scores as independent variables.**

| | Depressive temperament | | Cyclothymic temperament | | Irritable temperament | | Anxious temperament | | Hyperthymic temperament | |
|---|---|---|---|---|---|---|---|---|---|---|
| | Beta | $p$-value | Beta | $p$-value | Beta | $p$-value | Beta | $p$-value | Beta | $p$-value |
| Paternal care | −0.347 | 0.006 | −0.160 | 0.220 | −0.241 | 0.062 | 0.136 | 0.315 | −0.090 | 0.499 |
| Maternal care | 0.184 | 0.170 | −0.144 | 0.310 | −0.012 | 0.931 | 0.161 | 0.274 | −0.217 | 0.134 |
| Paternal Overprotection | 0.104 | 0.449 | 0.120 | 0.411 | −0.100 | 0.488 | 0.122 | 0.420 | 0.005 | 0.971 |
| Maternal Overprotection | 0.200 | 0.170 | −0.047 | 0.759 | 0.273 | 0.074 | 0.091 | 0.568 | −0.025 | 0.871 |
| Adjusted $R^2$ | 0.163 | | 0.062 | | 0.094 | | −0.007 | | 0.025 | |
| $p$-value | < 0.001 | | 0.041 | | 0.010 | | 0.507 | | 0.174 | |

Beta = standardized partial regression coefficient

Dependent variables: the scores of the five subscales of TEMPS-A (depressive temperament, cyclothymic temperament, irritable temperament, anxious temperament, and hyperthymic temperament)

Independent variables: four PBI subscale scores (paternal care, maternal care, paternal overprotection, and maternal overprotection)

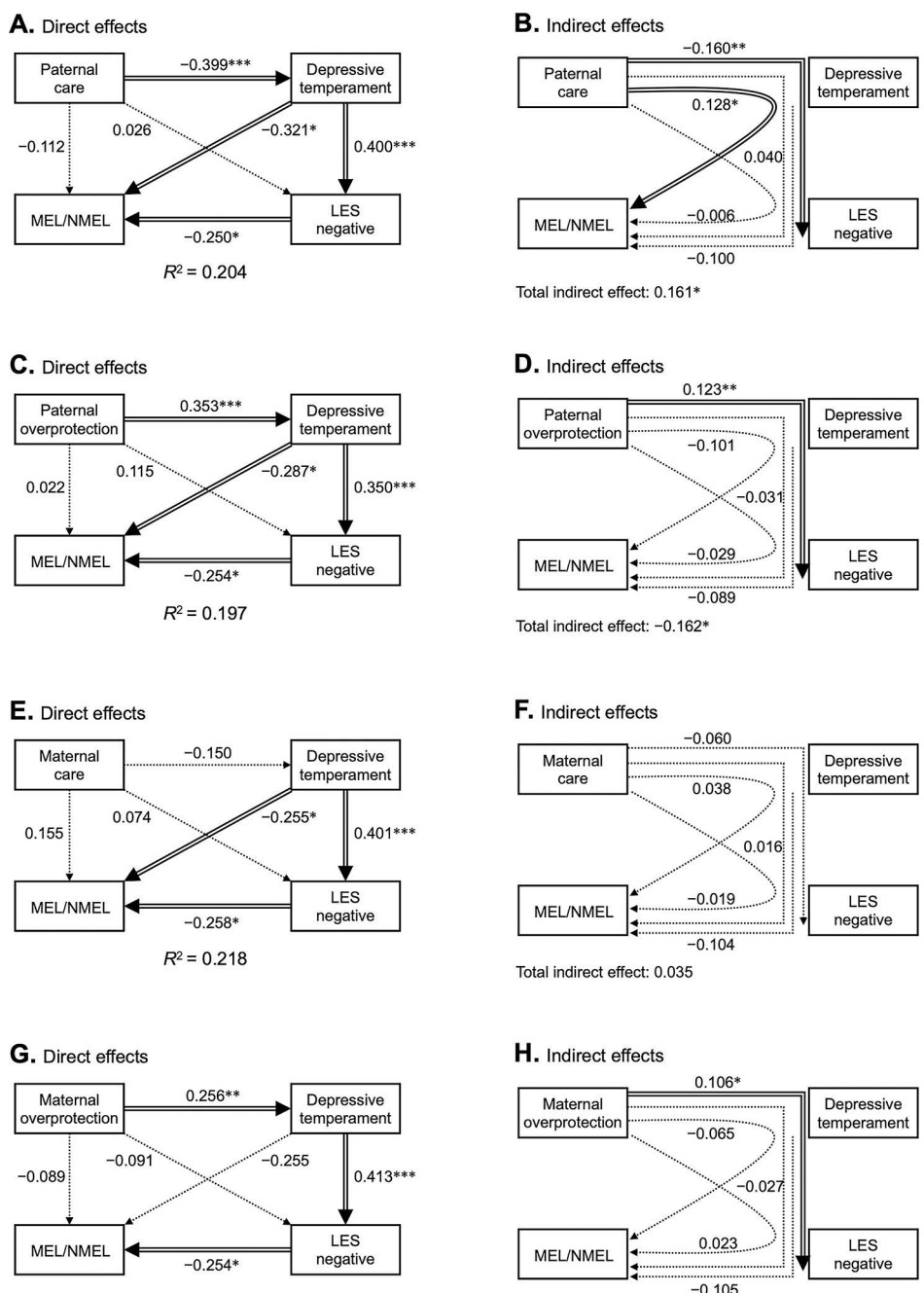

**Fig 1. Path analysis modeling the interrelationship between dysfunctional parenting, depressive temperament, stressful life events, and MEL/NMEL.** The results of direct (A, C, E, and G) and indirect (B, D, F, and H) effects of path analysis of 98 MDD patients with PBI subscales, depressive temperament, negative change score of LES, and the diagnosis of MEL (MEL = 1, NMEL = 0). The PBI subscales consist of paternal care (A and B), paternal overprotection (C and D), maternal care (E and F), and maternal overprotection (G and H), and the results of the path analysis of each subscale are shown. The double-lined arrows indicate the statistically significant pathways, and the dotted arrows indicate the nonsignificant pathways. The numbers beside the arrows represent the direct or indirect standardized coefficients. Abbreviations: LES, Life Experiences Survey; MDD, major depressive disorder; MEL, melancholic major depressive disorder; NMEL, non-melancholic major depressive disorder; PBI, Parental Bonding Instrument. $*p < 0.05$, $**p < 0.01$, $***p < 0.001$.

In the model shown in Fig 1A, paternal care had a significant negative direct effect on depressive temperament. Depressive temperament had a significant positive direct effect on the negative change score of the LES and a significant negative direct effect on the diagnosis of MEL. The negative change score of the LES showed a significant direct negative effect on the diagnosis of MEL. Paternal care had no significant direct effect on the diagnosis of MEL.

Fig 1B shows the indirect effects of the model including paternal care. Paternal care showed a significant indirect effect on the diagnosis of MEL via depressive temperament. The indirect path from paternal care to the negative change score of the LES via depressive temperament was also significant, but the other indirect paths were not significant. The total indirect effect of paternal care on the diagnosis of MEL was significant (indirect path coefficient = 0.161, $p < 0.05$).

In the model in Fig 1C, paternal overprotection had a significant direct positive effect on depressive temperament. Depressive temperament had a significant direct positive effect on the negative change scores of the LES, and a significant direct negative effect on the diagnosis of MEL. The negative change scores of the LES showed a significant direct negative effect on the diagnosis of MEL. Paternal overprotection had no significant direct effect on the diagnosis of MEL.

Fig 1D shows the indirect effects of the model, including paternal overprotection. Paternal overprotection had a significant total indirect effect on the diagnosis of MEL (indirect path coefficient = −0.162, $p < 0.05$). The indirect path from paternal overprotection to the negative change scores of the LES via depressive temperament was significant, but the other indirect paths were not significant.

In Fig 1E, maternal care had no significant direct effect on either depressive temperament, negative change scores of the LES, or diagnosis of MEL. Depressive temperament had a significant positive direct effect on the negative change scores of the LES, and a significant negative direct effect on the diagnosis of MEL. The negative change scores of the LES showed a significant negative direct effect on the diagnosis of MEL.

Fig 1F shows the indirect effects of the model including maternal care, but there were no significant indirect paths. The total indirect effect of maternal care on the diagnosis of MEL was not significant.

In Fig 1G, maternal overprotection had a significant direct positive effect on depressive temperament, but no significant direct effect on the negative change scores of the LES or the diagnosis of MEL. Depressive temperament had a significant direct positive effect on the negative change scores of the LES, but no significant direct effect on the diagnosis of MEL. The negative change scores of the LES had a significant direct negative effect on the diagnosis of MEL.

Fig 1H shows the indirect effects of the model including maternal overprotection. The indirect path from maternal overprotection to the negative change scores of the LES via depressive temperament was significant, whereas the other indirect paths were not significant. The total indirect effect of maternal overprotection on the diagnosis of MEL was not significant.

## Discussion

Our study demonstrated that patients with MDD who had received low paternal care in childhood were more likely to fit the diagnosis of NMEL than MEL, via increased depressive temperament. To the best of our knowledge, this is the first study to use path analysis to determine the interrelationship of dysfunctional parenting, affective temperaments, and stressful life events with the onset of MEL or NMEL. Traditionally, it has been suggested that the development of NMEL is associated with having experienced significant difficulties in childhood and adolescence in relation to one's parents [13]. However, our study indicated that dysfunctional parenting does not directly affect the development of NMEL, but rather indirectly through the mediation of depressive temperament.

The possibility that personality traits mediate the association between low levels of parental care and NMEL has been noted previously [14, 31]. However, the personality traits that mediate this association had remained unclear. Whiffen et al. [32] stated that low levels of parental care, mediated by personality dysfunction characterized by self-critical traits, affect the development of NMEL. The depressive temperament of the TEMPS-A and self-critical traits may be considered similar traits, as they share common characteristics of being unable to see the bright side of things and focusing on failure rather than success.

In this study, the total indirect effect from paternal overprotection to the diagnosis of MEL or NMEL was significant, but no significant indirect paths were identified. The indirect path of paternal overprotection to NMEL via depressive temperament had a *p*-value of 0.055 and may have been significant if the number of cases had been further increased. However, as there is presently no validated method of power testing for path analysis, it was not possible to estimate the sample size required to obtain statistically significant results.

Regarding the association between stressful life events and the diagnosis of MEL and NMEL, a direct association between NMEL and stressful life events was demonstrated, but the indirect path from dysfunctional parenting or depressive temperament to NMEL via stressful life events was not significant. The indirect path from paternal care to NMEL via depressive temperament and stressful life events had a *p*-value of 0.08, which may have also been significant if a larger sample size was analyzed. However, it can be said that dysfunctional parenting affects the development of NMEL through the mediation of affective temperament rather than stressful life events.

In this study, we found that the influence of paternal parenting was greater than maternal parenting in distinguishing between MEL and NMEL. Few studies to date have investigated how parenting styles influence the development of MEL and NMEL separately by the sex of the parents. In a previous study [15], NMEL patients were more likely to report abnormal parenting characteristics in the same-sex parent, but as there were almost an equal number of male and female patients in this study, this cannot be cited as a reason. Neale et al. [44] suggested in a study on twins that whereas mothers consistently treat both children with the same attitude, fathers may show different parenting attitudes depending on the child. It is hence possible that the father's parenting style is more variable than the mother's parenting style in the patients of this study.

In our sample, NMEL patients had more first-degree relatives with mood disorders than MEL patients. Traditionally, it is often assumed that MEL patients more often have a family history of mood disorders, but studies of DSM-defined melancholia have not supported this [45, 46]. According to Zimmerman et al. [46], among patients with mild depression who did not require hospitalization, relatives of NMEL patients had a significantly higher morbidity risk of depression than relatives of MEL patients. Thus, it is possible that there was a higher frequency of first-degree relatives with mood disorders in NMEL patients than in MEL patients, as our present study also included a large number of outpatients with relatively mild illnesses. In our present study, the number of patients whose fathers had mood disorders was the same in the MEL and NMEL groups (2 patients each), so differences in family history did not affect the results.

In clinical practice, physical therapy, including pharmacotherapy, is prioritized for MEL patients, whereas psychotherapy is considered important for NMEL patients. From the results of this study, a more effective approach to psychotherapy for NMEL patients may be possible if the involvement of paternal care and depressive temperament is taken into account. To rigorously investigate the interrelationship between the development of MEL and NMEL, dysfunctional parenting, affective temperaments, and stressful life events, a prospective study using a representative birth cohort is needed in the future. Additionally, increasing the sample size would enable the analysis to be performed in patient groups of each sex, which could lead

to the identification of even more important associations, including differences between the sexes.

There are several limitations to this study. First, the sample size was relatively small, thereby raising the possibility of a type II error. Second, this study was conducted using cross-sectional data. Although Parker [47] argued that the severity of depressive symptoms did not influence the PBI score, it cannot be ruled out that the current depressive state of the patient affected their evaluation on the LES and TEMPS-A. Third, there are some issues regarding the DSM-5 criteria for MEL. Criticisms have been raised that the DSM-5 criteria for MEL overlap with the diagnostic criteria for MDD, making it difficult to clearly distinguish between MEL and NMEL [11]. The use of alternative diagnostic criteria may lead to different results, and hence there are limitations to the generalization of the results of this study.

## Conclusion

We found that low levels of paternal care did not directly affect the development of NMEL, but affected the development of NMEL through the mediation of depressive temperament rather than stressful life events.

## Supporting information

**S1 Checklist. STROBE statement—checklist of items that should be included in reports of observational studies.**
(DOCX)

## Acknowledgments

We thank Dr. Helena Popiel of the Department of International Medical Communications, Tokyo Medical University, for editorial review of the manuscript.

## Author contributions

**Conceptualization:** Yu Tamada, Takeshi Inoue, Susumu Ohmae.

**Data curation:** Yu Tamada, Takeshi Inoue, Atsushi Sekine, Hiroyuki Toda, Minoru Takeshima, Masaaki Sasaki.

**Formal analysis:** Yu Tamada, Takeshi Inoue.

**Funding acquisition:** Yu Tamada.

**Investigation:** Yu Tamada, Atsushi Sekine, Hiroyuki Toda, Minoru Takeshima, Masaaki Sasaki.

**Methodology:** Yu Tamada, Takeshi Inoue.

**Project administration:** Yu Tamada.

**Supervision:** Takeshi Inoue, Yota Fujimura, Susumu Ohmae.

**Writing – original draft:** Yu Tamada.

**Writing – review & editing:** Takeshi Inoue, Atsushi Sekine, Hiroyuki Toda, Minoru Takeshima, Masaaki Sasaki, Yota Fujimura, Susumu Ohmae.

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
