## [Decision Letter · Decision Letter 0]

1 Sep 2023

PONE-D-23-16172Impact of dysfunctional parenting, affective temperaments, and stressful life events on the development of melancholic and non-melancholic depression: a path analysis studyPLOS ONEPONE-D-23-16172Impact of dysfunctional parenting, affective temperaments, and stressful life events on the development of melancholic and non-melancholic depression: a path analysis studyPLOS ONE

Dear Dr. Yu Tamada,Dear Dr. Yu Tamada,

Thank you for submitting your manuscript to PLOS ONE. After careful consideration, we feel that it has merit but does not fully meet PLOS ONE’s publication criteria as it currently stands. Therefore, we invite you to submit a revised version of the manuscript that addresses the points raised during the review process.Thank you for submitting your manuscript to PLOS ONE. After careful consideration, we feel that it has merit but does not fully meet PLOS ONE’s publication criteria as it currently stands. Therefore, we invite you to submit a revised version of the manuscript that addresses the points raised during the review process.

Please include the following items when submitting your revised manuscript:Please include the following items when submitting your revised manuscript:A rebuttal letter that responds to each point raised by the academic editor and reviewer(s). You should upload this letter as a separate file labeled 'Response to Reviewers'.A rebuttal letter that responds to each point raised by the academic editor and reviewer(s). You should upload this letter as a separate file labeled 'Response to Reviewers'.A marked-up copy of your manuscript that highlights changes made to the original version. You should upload this as a separate file labeled 'Revised Manuscript with Track Changes'.A marked-up copy of your manuscript that highlights changes made to the original version. You should upload this as a separate file labeled 'Revised Manuscript with Track Changes'.An unmarked version of your revised paper without tracked changes. You should upload this as a separate file labeled 'Manuscript'.An unmarked version of your revised paper without tracked changes. You should upload this as a separate file labeled 'Manuscript'.If applicable, we recommend that you deposit your laboratory protocols in protocols.io to enhance the reproducibility of your results. Protocols.io assigns your protocol its own identifier (DOI) so that it can be cited independently in the future. For instructions see: If applicable, we recommend that you deposit your laboratory protocols in protocols.io to enhance the reproducibility of your results. Protocols.io assigns your protocol its own identifier (DOI) so that it can be cited independently in the future. For instructions see: https://journals.plos.org/plosone/s/submission-guidelines#loc-laboratory-protocols. Additionally, PLOS ONE offers an option for publishing peer-reviewed Lab Protocol articles, which describe protocols hosted on protocols.io. Read more information on sharing protocols at . Additionally, PLOS ONE offers an option for publishing peer-reviewed Lab Protocol articles, which describe protocols hosted on protocols.io. Read more information on sharing protocols at https://plos.org/protocols?utm_medium=editorial-email&utm_source=authorletters&utm_campaign=protocols..

We look forward to receiving your revised manuscript.We look forward to receiving your revised manuscript.

Kind regards,Kind regards,

Yasuhiko Deguchi, M.D.,Ph.D.Yasuhiko Deguchi, M.D.,Ph.D.

Academic EditorAcademic Editor

PLOS ONEPLOS ONE

Journal Requirements:Journal Requirements:

When submitting your revision, we need you to address these additional requirements.When submitting your revision, we need you to address these additional requirements.

1. Please ensure that your manuscript meets PLOS ONE's style requirements, including those for file naming. The PLOS ONE style templates can be found at1. Please ensure that your manuscript meets PLOS ONE's style requirements, including those for file naming. The PLOS ONE style templates can be found at

https://journals.plos.org/plosone/s/file?id=wjVg/PLOSOne_formatting_sample_main_body.pdf andand

2 Thank you for stating the following in the Competing Interests section:2 Thank you for stating the following in the Competing Interests section:

"I have read the journal's policy and the authors of this manuscript have the following competing interests: YT received honoraria from Otsuka Pharmaceutical, Sumitomo Pharma, Eisai, MSD and Meiji Seika Pharma. TI has received personal compensation from Mochida Pharmaceutical, Takeda Pharmaceutical, Eli Lilly, Janssen Pharmaceutical, MSD, Taisho Toyama Pharmaceutical, Yoshitomiyakuhin, and Daiichi Sankyo; grants from Shionogi, Astellas, Tsumura, and Eisai; and grants and personal compensation from Otsuka Pharmaceutical, Sumitomo Pharma, Mitsubishi Tanabe Pharma, Kyowa Pharmaceutical Industry, Pfizer, Novartis Pharma, and Meiji Seika Pharma; and is a member of the advisory boards of Pfizer, Novartis Pharma, and Mitsubishi Tanabe Pharma. AS has received personal compensation from Otsuka Pharmaceutical, Meiji Seika Pharma, Takeda Pharmaceutical, and Sumitomo Pharma. HT has received lecture fees from Sumitomo Pharma, Otsuka Pharmaceutical, Takeda Pharmaceutical, Pfizer, Yoshitomi Pharmaceutical, and Viatris. MT has received lecture fees from Kyowa Pharmaceutical, Otsuka Pharmaceutical, Sumitomo Pharma, Takeda Pharmaceutical, and Lundbeck. MS has received lecture fees from Viatris. YF has received honoraria from Sumitomo Pharma, and research grants from Shionogi and Sumitomo Pharma. SO has received honoraria from Viatris and Takeda Pharmaceutical. This does not alter our adherence to PLOS ONE policies on sharing data and materials.""I have read the journal's policy and the authors of this manuscript have the following competing interests: YT received honoraria from Otsuka Pharmaceutical, Sumitomo Pharma, Eisai, MSD and Meiji Seika Pharma. TI has received personal compensation from Mochida Pharmaceutical, Takeda Pharmaceutical, Eli Lilly, Janssen Pharmaceutical, MSD, Taisho Toyama Pharmaceutical, Yoshitomiyakuhin, and Daiichi Sankyo; grants from Shionogi, Astellas, Tsumura, and Eisai; and grants and personal compensation from Otsuka Pharmaceutical, Sumitomo Pharma, Mitsubishi Tanabe Pharma, Kyowa Pharmaceutical Industry, Pfizer, Novartis Pharma, and Meiji Seika Pharma; and is a member of the advisory boards of Pfizer, Novartis Pharma, and Mitsubishi Tanabe Pharma. AS has received personal compensation from Otsuka Pharmaceutical, Meiji Seika Pharma, Takeda Pharmaceutical, and Sumitomo Pharma. HT has received lecture fees from Sumitomo Pharma, Otsuka Pharmaceutical, Takeda Pharmaceutical, Pfizer, Yoshitomi Pharmaceutical, and Viatris. MT has received lecture fees from Kyowa Pharmaceutical, Otsuka Pharmaceutical, Sumitomo Pharma, Takeda Pharmaceutical, and Lundbeck. MS has received lecture fees from Viatris. YF has received honoraria from Sumitomo Pharma, and research grants from Shionogi and Sumitomo Pharma. SO has received honoraria from Viatris and Takeda Pharmaceutical. This does not alter our adherence to PLOS ONE policies on sharing data and materials."

Please confirm that this does not alter your adherence to all PLOS ONE policies on sharing data and materials, by including the following statement: "This does not alter our adherence to  PLOS ONE policies on sharing data and materials.” (as detailed online in our guide for authors Please confirm that this does not alter your adherence to all PLOS ONE policies on sharing data and materials, by including the following statement: "This does not alter our adherence to  PLOS ONE policies on sharing data and materials.” (as detailed online in our guide for authors http://journals.plos.org/plosone/s/competing-interests).  If there are restrictions on sharing of data and/or materials, please state these. Please note that we cannot proceed with consideration of your article until this information has been declared.).  If there are restrictions on sharing of data and/or materials, please state these. Please note that we cannot proceed with consideration of your article until this information has been declared.

Please include your updated Competing Interests statement in your cover letter; we will change the online submission form on your behalf.Please include your updated Competing Interests statement in your cover letter; we will change the online submission form on your behalf.

3. In your Data Availability statement, you have not specified where the minimal data set underlying the results described in your manuscript can be found. PLOS defines a study's minimal data set as the underlying data used to reach the conclusions drawn in the manuscript and any additional data required to replicate the reported study findings in their entirety. All PLOS journals require that the minimal data set be made fully available. For more information about our data policy, please see 3. In your Data Availability statement, you have not specified where the minimal data set underlying the results described in your manuscript can be found. PLOS defines a study's minimal data set as the underlying data used to reach the conclusions drawn in the manuscript and any additional data required to replicate the reported study findings in their entirety. All PLOS journals require that the minimal data set be made fully available. For more information about our data policy, please see http://journals.plos.org/plosone/s/data-availability..

Upon re-submitting your revised manuscript, please upload your study’s minimal underlying data set as either Supporting Information files or to a stable, public repository and include the relevant URLs, DOIs, or accession numbers within your revised cover letter. For a list of acceptable repositories, please see Upon re-submitting your revised manuscript, please upload your study’s minimal underlying data set as either Supporting Information files or to a stable, public repository and include the relevant URLs, DOIs, or accession numbers within your revised cover letter. For a list of acceptable repositories, please see http://journals.plos.org/plosone/s/data-availability#loc-recommended-repositories. Any potentially identifying patient information must be fully anonymized.. Any potentially identifying patient information must be fully anonymized.

Important: If there are ethical or legal restrictions to sharing your data publicly, please explain these restrictions in detail. Please see our guidelines for more information on what we consider unacceptable restrictions to publicly sharing data: Important: If there are ethical or legal restrictions to sharing your data publicly, please explain these restrictions in detail. Please see our guidelines for more information on what we consider unacceptable restrictions to publicly sharing data: http://journals.plos.org/plosone/s/data-availability#loc-unacceptable-data-access-restrictions. Note that it is not acceptable for the authors to be the sole named individuals responsible for ensuring data access.. Note that it is not acceptable for the authors to be the sole named individuals responsible for ensuring data access.

We will update your Data Availability statement to reflect the information you provide in your cover letter.We will update your Data Availability statement to reflect the information you provide in your cover letter.

4. Please include captions for your Supporting Information files at the end of your manuscript, and update any in-text citations to match accordingly. Please see our Supporting Information guidelines for more information: 4. Please include captions for your Supporting Information files at the end of your manuscript, and update any in-text citations to match accordingly. Please see our Supporting Information guidelines for more information: http://journals.plos.org/plosone/s/supporting-information..

5. Please review your reference list to ensure that it is complete and correct. If you have cited papers that have been retracted, please include the rationale for doing so in the manuscript text, or remove these references and replace them with relevant current references. Any changes to the reference list should be mentioned in the rebuttal letter that accompanies your revised manuscript. If you need to cite a retracted article, indicate the article’s retracted status in the References list and also include a citation and full reference for the retraction notice.5. Please review your reference list to ensure that it is complete and correct. If you have cited papers that have been retracted, please include the rationale for doing so in the manuscript text, or remove these references and replace them with relevant current references. Any changes to the reference list should be mentioned in the rebuttal letter that accompanies your revised manuscript. If you need to cite a retracted article, indicate the article’s retracted status in the References list and also include a citation and full reference for the retraction notice.

Reviewers' comments:Reviewers' comments:

Reviewer's Responses to QuestionsReviewer's Responses to Questions

**Comments to the Author**

1. Is the manuscript technically sound, and do the data support the conclusions?1. Is the manuscript technically sound, and do the data support the conclusions?

The manuscript must describe a technically sound piece of scientific research with data that supports the conclusions. Experiments must have been conducted rigorously, with appropriate controls, replication, and sample sizes. The conclusions must be drawn appropriately based on the data presented. The manuscript must describe a technically sound piece of scientific research with data that supports the conclusions. Experiments must have been conducted rigorously, with appropriate controls, replication, and sample sizes. The conclusions must be drawn appropriately based on the data presented.

Reviewer #1: YesReviewer #1: Yes

Reviewer #2: YesReviewer #2: Yes

2. Has the statistical analysis been performed appropriately and rigorously? 2. Has the statistical analysis been performed appropriately and rigorously?

Reviewer #1: YesReviewer #1: Yes

Reviewer #2: YesReviewer #2: Yes

3. Have the authors made all data underlying the findings in their manuscript fully available?3. Have the authors made all data underlying the findings in their manuscript fully available?

The The PLOS Data policy requires authors to make all data underlying the findings described in their manuscript fully available without restriction, with rare exception (please refer to the Data Availability Statement in the manuscript PDF file). The data should be provided as part of the manuscript or its supporting information, or deposited to a public repository. For example, in addition to summary statistics, the data points behind means, medians and variance measures should be available. If there are restrictions on publicly sharing data—e.g. participant privacy or use of data from a third party—those must be specified.requires authors to make all data underlying the findings described in their manuscript fully available without restriction, with rare exception (please refer to the Data Availability Statement in the manuscript PDF file). The data should be provided as part of the manuscript or its supporting information, or deposited to a public repository. For example, in addition to summary statistics, the data points behind means, medians and variance measures should be available. If there are restrictions on publicly sharing data—e.g. participant privacy or use of data from a third party—those must be specified.

Reviewer #1: NoReviewer #1: No

Reviewer #2: YesReviewer #2: Yes

4. Is the manuscript presented in an intelligible fashion and written in standard English?4. Is the manuscript presented in an intelligible fashion and written in standard English?

PLOS ONE does not copyedit accepted manuscripts, so the language in submitted articles must be clear, correct, and unambiguous. Any typographical or grammatical errors should be corrected at revision, so please note any specific errors here.PLOS ONE does not copyedit accepted manuscripts, so the language in submitted articles must be clear, correct, and unambiguous. Any typographical or grammatical errors should be corrected at revision, so please note any specific errors here.

Reviewer #1: YesReviewer #1: Yes

Reviewer #2: YesReviewer #2: Yes

5. Review Comments to the Author5. Review Comments to the Author

Please use the space provided to explain your answers to the questions above. You may also include additional comments for the author, including concerns about dual publication, research ethics, or publication ethics. (Please upload your review as an attachment if it exceeds 20,000 characters)Please use the space provided to explain your answers to the questions above. You may also include additional comments for the author, including concerns about dual publication, research ethics, or publication ethics. (Please upload your review as an attachment if it exceeds 20,000 characters)

Reviewer #1: The authors investigated to clarify the interrelationship between dysfunctional parenting, affective temperaments, stressful life events, and the diagnosis of melancholic depression and non-melancholic depression using path analysis. The authors demonstrated patients with major depressive disorder who had received low paternal care in childhood were more likely to fit the diagnosis of non-melancholic depression than melancholic depression via increased depressive temperament. This paper is well written. The study is quite exciting and significantly adds to the literature of the field. The following points should be addressed before publication.Reviewer #1: The authors investigated to clarify the interrelationship between dysfunctional parenting, affective temperaments, stressful life events, and the diagnosis of melancholic depression and non-melancholic depression using path analysis. The authors demonstrated patients with major depressive disorder who had received low paternal care in childhood were more likely to fit the diagnosis of non-melancholic depression than melancholic depression via increased depressive temperament. This paper is well written. The study is quite exciting and significantly adds to the literature of the field. The following points should be addressed before publication.

Minor commentsMinor comments

1) Line 1511) Line 151

Please describe the specific period time of “a sufficient duration”.Please describe the specific period time of “a sufficient duration”.

2) Line 193 Data analysis2) Line 193 Data analysis

Please mention that no correction for multiple testing was performed.Please mention that no correction for multiple testing was performed.

3) Line 399 and Line 4063) Line 399 and Line 406

Please consider conducting a power analysis to estimate the sample size to obtain statistically significant results.Please consider conducting a power analysis to estimate the sample size to obtain statistically significant results.

4) Table 1: Comorbid psychiatric disorder4) Table 1: Comorbid psychiatric disorder

Please provide information on what kind of psychiatric disorders are comorbid with melancholic depression and non-melancholic depression groups elsewhere in your paper.Please provide information on what kind of psychiatric disorders are comorbid with melancholic depression and non-melancholic depression groups elsewhere in your paper.

5) Table 1: First-degree relative with a mood disorder: n(%)5) Table 1: First-degree relative with a mood disorder: n(%)

Interestingly the “First-degree relative with a mood disorder: n(%)” is higher in non-melancholic depression than in melancholic depression. Please conduct a literature search on this and discuss whether it may affect the results of the current paper.Interestingly the “First-degree relative with a mood disorder: n(%)” is higher in non-melancholic depression than in melancholic depression. Please conduct a literature search on this and discuss whether it may affect the results of the current paper.

Reviewer #2: The authors showed that low levels of paternal care did not directly affect the development of NMEL, but affected the development of NMEL through the mediation of depressive temperament rather than stressful life events. This is interesting and worthy of publication.Reviewer #2: The authors showed that low levels of paternal care did not directly affect the development of NMEL, but affected the development of NMEL through the mediation of depressive temperament rather than stressful life events. This is interesting and worthy of publication.

6. PLOS authors have the option to publish the peer review history of their article (6. PLOS authors have the option to publish the peer review history of their article (what does this mean?). If published, this will include your full peer review and any attached files.). If published, this will include your full peer review and any attached files.

.

Reviewer #1: NoReviewer #1: No

Reviewer #2: NoReviewer #2: No

While revising your submission, please upload your figure files to the Preflight Analysis and Conversion Engine (PACE) digital diagnostic tool, While revising your submission, please upload your figure files to the Preflight Analysis and Conversion Engine (PACE) digital diagnostic tool, https://pacev2.apexcovantage.com/. PACE helps ensure that figures meet PLOS requirements. To use PACE, you must first register as a user. Registration is free. Then, login and navigate to the UPLOAD tab, where you will find detailed instructions on how to use the tool. If you encounter any issues or have any questions when using PACE, please email PLOS at . PACE helps ensure that figures meet PLOS requirements. To use PACE, you must first register as a user. Registration is free. Then, login and navigate to the UPLOAD tab, where you will find detailed instructions on how to use the tool. If you encounter any issues or have any questions when using PACE, please email PLOS at figures@plos.org. Please note that Supporting Information files do not need this step.. Please note that Supporting Information files do not need this step.

---

## [Author Response · Author response to Decision Letter 0]

28 Sep 2023

POINT-BY-POINT RESPONSES TO THE REVIEWERS’ COMMENTSPOINT-BY-POINT RESPONSES TO THE REVIEWERS’ COMMENTS

We thank the reviewers for the helpful advice and comments, which have enabled us to substantially improve our manuscript.We thank the reviewers for the helpful advice and comments, which have enabled us to substantially improve our manuscript.

Journal Requirements:Journal Requirements:

1. Please ensure that your manuscript meets PLOS ONE's style requirements, including those for file naming.1. Please ensure that your manuscript meets PLOS ONE's style requirements, including those for file naming.

Response:Response:

Thank you for the comment. We modified the manuscript style in accordance with the style requirements of PLOS ONE.Thank you for the comment. We modified the manuscript style in accordance with the style requirements of PLOS ONE.

2. Thank you for stating the following in the Competing Interests section. Please confirm that this does not alter your adherence to all PLOS ONE policies on sharing data and materials, by including the following statement: "This does not alter our adherence to PLOS ONE policies on sharing data and materials.” If there are restrictions on sharing of data and/or materials, please state these. Please note that we cannot proceed with consideration of your article until this information has been declared.2. Thank you for stating the following in the Competing Interests section. Please confirm that this does not alter your adherence to all PLOS ONE policies on sharing data and materials, by including the following statement: "This does not alter our adherence to PLOS ONE policies on sharing data and materials.” If there are restrictions on sharing of data and/or materials, please state these. Please note that we cannot proceed with consideration of your article until this information has been declared.

Response:Response:

Our original submission already included the statement “This does not alter our adherence to PLOS ONE policies on sharing data and materials.” in the ‘Competing Interests’ section. Therefore, we do not think any change is necessary, but we apologize if we have not understood your explanation correctly. Restrictions on data sharing are specified in the ‘Data Availability Statement’ section.Our original submission already included the statement “This does not alter our adherence to PLOS ONE policies on sharing data and materials.” in the ‘Competing Interests’ section. Therefore, we do not think any change is necessary, but we apologize if we have not understood your explanation correctly. Restrictions on data sharing are specified in the ‘Data Availability Statement’ section.

3. In your Data Availability statement, you have not specified where the minimal data set underlying the results described in your manuscript can be found. PLOS defines a study's minimal data set as the underlying data used to reach the conclusions drawn in the manuscript and any additional data required to replicate the reported study findings in their entirety.3. In your Data Availability statement, you have not specified where the minimal data set underlying the results described in your manuscript can be found. PLOS defines a study's minimal data set as the underlying data used to reach the conclusions drawn in the manuscript and any additional data required to replicate the reported study findings in their entirety.

Response:Response:

Thank you for your comment. We have changed the Data Availability statement, as follows.Thank you for your comment. We have changed the Data Availability statement, as follows.

“Data Availability Statement: Data cannot be shared publicly because of Ethics Committee restriction. Data are available from the Internal Review Board of Toranomon Hospital (Japan) (contact via email: “Data Availability Statement: Data cannot be shared publicly because of Ethics Committee restriction. Data are available from the Internal Review Board of Toranomon Hospital (Japan) (contact via email: ytmd@tokyo-med.ac.jp) for researchers who meet the criteria for access to confidential data.”) for researchers who meet the criteria for access to confidential data.”

4. Please include captions for your Supporting Information files at the end of your manuscript, and update any in-text citations to match accordingly.4. Please include captions for your Supporting Information files at the end of your manuscript, and update any in-text citations to match accordingly.

Response:Response:

Thank you for the comments. There are no Supporting Information files in this paper.Thank you for the comments. There are no Supporting Information files in this paper.

5. Please review your reference list to ensure that it is complete and correct. If you have cited papers that have been retracted, please include the rationale for doing so in the manuscript text, or remove these references and replace them with relevant current references. Any changes to the reference list should be mentioned in the rebuttal letter that accompanies your revised manuscript. If you need to cite a retracted article, indicate the article’s retracted status in the References list and also include a citation and full reference for the retraction notice.5. Please review your reference list to ensure that it is complete and correct. If you have cited papers that have been retracted, please include the rationale for doing so in the manuscript text, or remove these references and replace them with relevant current references. Any changes to the reference list should be mentioned in the rebuttal letter that accompanies your revised manuscript. If you need to cite a retracted article, indicate the article’s retracted status in the References list and also include a citation and full reference for the retraction notice.

Response:Response:

We thank you for the comments. We confirmed that all references are correct and that we have not included any retracted papers. In addition, the following references have been added as a result of manuscript revision, and the bibliography numbers have been adjusted to reflect the changes made to the manuscript.We thank you for the comments. We confirmed that all references are correct and that we have not included any retracted papers. In addition, the following references have been added as a result of manuscript revision, and the bibliography numbers have been adjusted to reflect the changes made to the manuscript.

45. McGrath PJ, Khan AY, Trivedi MH, Stewart JW, Morris DW, Wisniewski SR, et al. Response to a selective serotonin reuptake inhibitor (citalopram) in major depressive disorder with melancholic features: a STAR*D report. J Clin Psychiatry. 2008;69(12):1847-55. Epub 20081118. doi: 10.4088/jcp.v69n1201. PubMed PMID: 19026268.45. McGrath PJ, Khan AY, Trivedi MH, Stewart JW, Morris DW, Wisniewski SR, et al. Response to a selective serotonin reuptake inhibitor (citalopram) in major depressive disorder with melancholic features: a STAR*D report. J Clin Psychiatry. 2008;69(12):1847-55. Epub 20081118. doi: 10.4088/jcp.v69n1201. PubMed PMID: 19026268.

46. Zimmerman M, Black DW, Coryell W. Diagnostic criteria for melancholia. The comparative validity of DSM-III and DSM-III-R. Arch Gen Psychiatry. 1989;46(4):361-8. doi: 10.1001/archpsyc.1989.01810040067010. PubMed PMID: 2930332.46. Zimmerman M, Black DW, Coryell W. Diagnostic criteria for melancholia. The comparative validity of DSM-III and DSM-III-R. Arch Gen Psychiatry. 1989;46(4):361-8. doi: 10.1001/archpsyc.1989.01810040067010. PubMed PMID: 2930332.

REVIEWER #1REVIEWER #1

1. Line 151: Please describe the specific period time of “a sufficient duration”.1. Line 151: Please describe the specific period time of “a sufficient duration”.

Response:Response:

We thank you for your comment. In accordance with your suggestion, we have modified the description as follows.We thank you for your comment. In accordance with your suggestion, we have modified the description as follows.

“The definition of treatment resistance was the absence of a response to at least 2 antidepressants from different pharmacological classes that were administered for at least 4 consecutive weeks.” (page 8, lines 149 to 151)“The definition of treatment resistance was the absence of a response to at least 2 antidepressants from different pharmacological classes that were administered for at least 4 consecutive weeks.” (page 8, lines 149 to 151)

2. Line 193 Data analysis: Please mention that no correction for multiple testing was performed.2. Line 193 Data analysis: Please mention that no correction for multiple testing was performed.

Response:Response:

Thank you for your important comment. In accordance with the comment, we have added that no correction for multiple testing was performed, as follows.Thank you for your important comment. In accordance with the comment, we have added that no correction for multiple testing was performed, as follows.

“No correction for multiple testing was performed.” (page 10, line 214)“No correction for multiple testing was performed.” (page 10, line 214)

3. Line 399 and Line 406: Please consider conducting a power analysis to estimate the sample size to obtain statistically significant results.3. Line 399 and Line 406: Please consider conducting a power analysis to estimate the sample size to obtain statistically significant results.

Response:Response:

Thank you for your important comment. We have consulted with our statisticians, but unfortunately, we are unable to perform a power analysis because at present there is no validated method of power testing for path analysis. Therefore, we have added the following statement to the Discussion section of the revised manuscript.Thank you for your important comment. We have consulted with our statisticians, but unfortunately, we are unable to perform a power analysis because at present there is no validated method of power testing for path analysis. Therefore, we have added the following statement to the Discussion section of the revised manuscript.

“However, as there is presently no validated method of power testing for path analysis, it was not possible to estimate the sample size required to obtain statistically significant results.” (pages 20 to 21, lines 403 to 405)“However, as there is presently no validated method of power testing for path analysis, it was not possible to estimate the sample size required to obtain statistically significant results.” (pages 20 to 21, lines 403 to 405)

4. Table 1: Comorbid psychiatric disorder: Please provide information on what kind of psychiatric disorders are comorbid with melancholic depression and non-melancholic depression groups elsewhere in your paper.4. Table 1: Comorbid psychiatric disorder: Please provide information on what kind of psychiatric disorders are comorbid with melancholic depression and non-melancholic depression groups elsewhere in your paper.

Response:Response:

Thank you for your comment. In accordance with your comments, we have added information on comorbid psychiatric disorders to the Results section of the revised manuscript, as follows.Thank you for your comment. In accordance with your comments, we have added information on comorbid psychiatric disorders to the Results section of the revised manuscript, as follows.

“Comorbid psychiatric disorders included panic disorder in 1 patient in the MEL group, and autism spectrum disorder in 3 patients, and social anxiety disorder, generalized anxiety disorder, panic disorder, and borderline personality disorder in 1 patient each in the NMEL group.” (page 11, lines 224 to 227)“Comorbid psychiatric disorders included panic disorder in 1 patient in the MEL group, and autism spectrum disorder in 3 patients, and social anxiety disorder, generalized anxiety disorder, panic disorder, and borderline personality disorder in 1 patient each in the NMEL group.” (page 11, lines 224 to 227)

5. Table 1: First-degree relative with a mood disorder: n(%): Interestingly the “First-degree relative with a mood disorder: n(%)” is higher in non-melancholic depression than in melancholic depression. Please conduct a literature search on this and discuss whether it may affect the results of the current paper.5. Table 1: First-degree relative with a mood disorder: n(%): Interestingly the “First-degree relative with a mood disorder: n(%)” is higher in non-melancholic depression than in melancholic depression. Please conduct a literature search on this and discuss whether it may affect the results of the current paper.

Response:Response:

Thank you for your important comment. In accordance with your comment, we have added the following discussion of differences in family history of the patients to the revised manuscript.Thank you for your important comment. In accordance with your comment, we have added the following discussion of differences in family history of the patients to the revised manuscript.

“In our sample, NMEL patients had more first-degree relatives with mood disorders than MEL patients. Traditionally, it is often assumed that MEL patients more often have a family history of mood disorders, but studies of DSM-defined melancholia have not supported this [“In our sample, NMEL patients had more first-degree relatives with mood disorders than MEL patients. Traditionally, it is often assumed that MEL patients more often have a family history of mood disorders, but studies of DSM-defined melancholia have not supported this [[45], , [46]]. According to Zimmerman et al. [46], among patients with mild depression who did not require hospitalization, relatives of NMEL patients had a significantly higher morbidity risk of depression than relatives of MEL patients. Thus, it is possible that there was a higher frequency of first-degree relatives with mood disorders in NMEL patients than in MEL patients, as our present study also included a large number of outpatients with relatively mild illnesses. In our present study, the number of patients whose fathers had mood disorders was the same in the MEL and NMEL groups (2 patients each), so differences in family history did not affect the results.” (pages 21 to 22, lines 424 to 434)], among patients with mild depression who did not require hospitalization, relatives of NMEL patients had a significantly higher morbidity risk of depression than relatives of MEL patients. Thus, it is possible that there was a higher frequency of first-degree relatives with mood disorders in NMEL patients than in MEL patients, as our present study also included a large number of outpatients with relatively mild illnesses. In our present study, the number of patients whose fathers had mood disorders was the same in the MEL and NMEL groups (2 patients each), so differences in family history did not affect the results.” (pages 21 to 22, lines 424 to 434)

We have also added 2 newly referenced papers to the References section.We have also added 2 newly referenced papers to the References section.

“45. McGrath PJ, Khan AY, Trivedi MH, Stewart JW, Morris DW, Wisniewski SR, et al. Response to a selective serotonin reuptake inhibitor (citalopram) in major depressive disorder with melancholic features: a STAR*D report. J Clin Psychiatry. 2008;69(12):1847-55. Epub 20081118. doi: 10.4088/jcp.v69n1201. PubMed PMID: 19026268.“45. McGrath PJ, Khan AY, Trivedi MH, Stewart JW, Morris DW, Wisniewski SR, et al. Response to a selective serotonin reuptake inhibitor (citalopram) in major depressive disorder with melancholic features: a STAR*D report. J Clin Psychiatry. 2008;69(12):1847-55. Epub 20081118. doi: 10.4088/jcp.v69n1201. PubMed PMID: 19026268.

46. Zimmerman M, Black DW, Coryell W. Diagnostic criteria for melancholia. The comparative validity of DSM-III and DSM-III-R. Arch Gen Psychiatry. 1989;46(4):361-8. doi: 10.1001/archpsyc.1989.01810040067010. PubMed PMID: 2930332.” (page 31, lines 616 to 624)46. Zimmerman M, Black DW, Coryell W. Diagnostic criteria for melancholia. The comparative validity of DSM-III and DSM-III-R. Arch Gen Psychiatry. 1989;46(4):361-8. doi: 10.1001/archpsyc.1989.01810040067010. PubMed PMID: 2930332.” (page 31, lines 616 to 624)

---

## [Decision Letter · Decision Letter 1]

25 Oct 2023

Impact of dysfunctional parenting, affective temperaments, and stressful life events on the development of melancholic and non-melancholic depression: a path analysis studyImpact of dysfunctional parenting, affective temperaments, and stressful life events on the development of melancholic and non-melancholic depression: a path analysis study

PONE-D-23-16172R1PONE-D-23-16172R1

Dear Dr. Yu Tamada,Dear Dr. Yu Tamada,

We’re pleased to inform you that your manuscript has been judged scientifically suitable for publication and will be formally accepted for publication once it meets all outstanding technical requirements.We’re pleased to inform you that your manuscript has been judged scientifically suitable for publication and will be formally accepted for publication once it meets all outstanding technical requirements.

Within one week, you’ll receive an e-mail detailing the required amendments. When these have been addressed, you’ll receive a formal acceptance letter and your manuscript will be scheduled for publication.Within one week, you’ll receive an e-mail detailing the required amendments. When these have been addressed, you’ll receive a formal acceptance letter and your manuscript will be scheduled for publication.

An invoice for payment will follow shortly after the formal acceptance. To ensure an efficient process, please log into Editorial Manager at An invoice for payment will follow shortly after the formal acceptance. To ensure an efficient process, please log into Editorial Manager at http://www.editorialmanager.com/pone/, click the 'Update My Information' link at the top of the page, and double check that your user information is up-to-date. If you have any billing related questions, please contact our Author Billing department directly at , click the 'Update My Information' link at the top of the page, and double check that your user information is up-to-date. If you have any billing related questions, please contact our Author Billing department directly at authorbilling@plos.org..

If your institution or institutions have a press office, please notify them about your upcoming paper to help maximize its impact. If they’ll be preparing press materials, please inform our press team as soon as possible -- no later than 48 hours after receiving the formal acceptance. Your manuscript will remain under strict press embargo until 2 pm Eastern Time on the date of publication. For more information, please contact If your institution or institutions have a press office, please notify them about your upcoming paper to help maximize its impact. If they’ll be preparing press materials, please inform our press team as soon as possible -- no later than 48 hours after receiving the formal acceptance. Your manuscript will remain under strict press embargo until 2 pm Eastern Time on the date of publication. For more information, please contact onepress@plos.org..

Kind regards,Kind regards,

Yasuhiko Deguchi, M.D.,Ph.D.Yasuhiko Deguchi, M.D.,Ph.D.

Academic EditorAcademic Editor

PLOS ONEPLOS ONE

Additional Editor Comments (optional):Additional Editor Comments (optional):

Reviewers' comments:Reviewers' comments:

Reviewer's Responses to QuestionsReviewer's Responses to Questions

**Comments to the Author**

1. If the authors have adequately addressed your comments raised in a previous round of review and you feel that this manuscript is now acceptable for publication, you may indicate that here to bypass the “Comments to the Author” section, enter your conflict of interest statement in the “Confidential to Editor” section, and submit your "Accept" recommendation.1. If the authors have adequately addressed your comments raised in a previous round of review and you feel that this manuscript is now acceptable for publication, you may indicate that here to bypass the “Comments to the Author” section, enter your conflict of interest statement in the “Confidential to Editor” section, and submit your "Accept" recommendation.

Reviewer #1: All comments have been addressedReviewer #1: All comments have been addressed

2. Is the manuscript technically sound, and do the data support the conclusions?2. Is the manuscript technically sound, and do the data support the conclusions?

The manuscript must describe a technically sound piece of scientific research with data that supports the conclusions. Experiments must have been conducted rigorously, with appropriate controls, replication, and sample sizes. The conclusions must be drawn appropriately based on the data presented. The manuscript must describe a technically sound piece of scientific research with data that supports the conclusions. Experiments must have been conducted rigorously, with appropriate controls, replication, and sample sizes. The conclusions must be drawn appropriately based on the data presented.

Reviewer #1: YesReviewer #1: Yes

3. Has the statistical analysis been performed appropriately and rigorously? 3. Has the statistical analysis been performed appropriately and rigorously?

Reviewer #1: YesReviewer #1: Yes

4. Have the authors made all data underlying the findings in their manuscript fully available?4. Have the authors made all data underlying the findings in their manuscript fully available?

The The PLOS Data policy requires authors to make all data underlying the findings described in their manuscript fully available without restriction, with rare exception (please refer to the Data Availability Statement in the manuscript PDF file). The data should be provided as part of the manuscript or its supporting information, or deposited to a public repository. For example, in addition to summary statistics, the data points behind means, medians and variance measures should be available. If there are restrictions on publicly sharing data—e.g. participant privacy or use of data from a third party—those must be specified.requires authors to make all data underlying the findings described in their manuscript fully available without restriction, with rare exception (please refer to the Data Availability Statement in the manuscript PDF file). The data should be provided as part of the manuscript or its supporting information, or deposited to a public repository. For example, in addition to summary statistics, the data points behind means, medians and variance measures should be available. If there are restrictions on publicly sharing data—e.g. participant privacy or use of data from a third party—those must be specified.

Reviewer #1: YesReviewer #1: Yes

5. Is the manuscript presented in an intelligible fashion and written in standard English?5. Is the manuscript presented in an intelligible fashion and written in standard English?

PLOS ONE does not copyedit accepted manuscripts, so the language in submitted articles must be clear, correct, and unambiguous. Any typographical or grammatical errors should be corrected at revision, so please note any specific errors here.PLOS ONE does not copyedit accepted manuscripts, so the language in submitted articles must be clear, correct, and unambiguous. Any typographical or grammatical errors should be corrected at revision, so please note any specific errors here.

Reviewer #1: YesReviewer #1: Yes

6. Review Comments to the Author6. Review Comments to the Author

Please use the space provided to explain your answers to the questions above. You may also include additional comments for the author, including concerns about dual publication, research ethics, or publication ethics. (Please upload your review as an attachment if it exceeds 20,000 characters)Please use the space provided to explain your answers to the questions above. You may also include additional comments for the author, including concerns about dual publication, research ethics, or publication ethics. (Please upload your review as an attachment if it exceeds 20,000 characters)

Reviewer #1: The authors investigated to clarify the interrelationship between dysfunctional parenting, affective temperaments, stressful life events, and the diagnosis of melancholic depression and non-melancholic depression using path analysis.Reviewer #1: The authors investigated to clarify the interrelationship between dysfunctional parenting, affective temperaments, stressful life events, and the diagnosis of melancholic depression and non-melancholic depression using path analysis.

The manuscript has been much improved and is in a nice condition now.The manuscript has been much improved and is in a nice condition now.

7. PLOS authors have the option to publish the peer review history of their article (7. PLOS authors have the option to publish the peer review history of their article (what does this mean?). If published, this will include your full peer review and any attached files.). If published, this will include your full peer review and any attached files.

.

Reviewer #1: Reviewer #1: **Yes:** Yuki KageyamaYuki Kageyama

---

## [Editor Report · Acceptance letter]

27 Oct 2023

PONE-D-23-16172R1 PONE-D-23-16172R1

Impact of dysfunctional parenting, affective temperaments, and stressful life events on the development of melancholic and non-melancholic depression: a path analysis study Impact of dysfunctional parenting, affective temperaments, and stressful life events on the development of melancholic and non-melancholic depression: a path analysis study

Dear Dr. Tamada:Dear Dr. Tamada:

I'm pleased to inform you that your manuscript has been deemed suitable for publication in PLOS ONE. Congratulations! Your manuscript is now with our production department. I'm pleased to inform you that your manuscript has been deemed suitable for publication in PLOS ONE. Congratulations! Your manuscript is now with our production department.

If your institution or institutions have a press office, please let them know about your upcoming paper now to help maximize its impact. If they'll be preparing press materials, please inform our press team within the next 48 hours. Your manuscript will remain under strict press embargo until 2 pm Eastern Time on the date of publication. For more information please contact If your institution or institutions have a press office, please let them know about your upcoming paper now to help maximize its impact. If they'll be preparing press materials, please inform our press team within the next 48 hours. Your manuscript will remain under strict press embargo until 2 pm Eastern Time on the date of publication. For more information please contact onepress@plos.org..

If we can help with anything else, please email us at If we can help with anything else, please email us at plosone@plos.org. .

Thank you for submitting your work to PLOS ONE and supporting open access. Thank you for submitting your work to PLOS ONE and supporting open access.

Kind regards, Kind regards,

PLOS ONE Editorial Office StaffPLOS ONE Editorial Office Staff

on behalf ofon behalf of

Dr. Yasuhiko Deguchi Dr. Yasuhiko Deguchi

Academic EditorAcademic Editor

PLOS ONEPLOS ONE